# Advances and Challenges in Spatial Transcriptomics for Developmental Biology

**DOI:** 10.3390/biom13010156

**Published:** 2023-01-12

**Authors:** Kyongho Choe, Unil Pak, Yu Pang, Wanjun Hao, Xiuqin Yang

**Affiliations:** 1College of Animal Science and Technology, Northeast Agricultural University, Harbin 150030, China; 2College of Landscape Architecture, Northeast Forestry University, Harbin 150040, China

**Keywords:** developmental biology, scRNA-seq, spatial resolution, spatial transcriptomic

## Abstract

Development from single cells to multicellular tissues and organs involves more than just the exact replication of cells, which is known as differentiation. The primary focus of research into the mechanism of differentiation has been differences in gene expression profiles between individual cells. However, it has predominantly been conducted at low throughput and bulk levels, challenging the efforts to understand molecular mechanisms of differentiation during the developmental process in animals and humans. During the last decades, rapid methodological advancements in genomics facilitated the ability to study developmental processes at a genome-wide level and finer resolution. Particularly, sequencing transcriptomes at single-cell resolution, enabled by single-cell RNA-sequencing (scRNA-seq), was a breath-taking innovation, allowing scientists to gain a better understanding of differentiation and cell lineage during the developmental process. However, single-cell isolation during scRNA-seq results in the loss of the spatial information of individual cells and consequently limits our understanding of the specific functions of the cells performed by different spatial regions of tissues or organs. This greatly encourages the emergence of the spatial transcriptomic discipline and tools. Here, we summarize the recent application of scRNA-seq and spatial transcriptomic tools for developmental biology. We also discuss the limitations of current spatial transcriptomic tools and approaches, as well as possible solutions and future prospects.

## 1. Introduction

The process by which a complex multicellular organism grows from a single cell is the subject of developmental biology. Multicellular organisms are not born completely developed. Instead, they emerge from a single cell through a rather slow, progressive changing process that we refer to as development. In the past decades, which genes are expressed, where and when they are expressed, and at what degree of expression are fundamental concerns in developmental biology. These concerns have initially been addressed by studying one gene at a time, but a new area of study that deals with multiple genes concurrently has developed. Thanks to a methodology that has been discovered at the interface of large-scale genomic approaches and developmental biology, developmental biologists can now generate information about the large number of genes and their pathways, which can provide an integrated view of complex developmental processes [1]. Among the large-scale genomic approaches, RNA sequencing (RNA-seq) is the method that has received the most attention. As RNA serves multiple functions as a messenger, regulatory molecule, or critical component of all intracellular biological processes, RNA-seq can provide all necessary information for cellular activities [2]. Nonetheless, conventional RNA-seq (called bulk RNA-seq) had been conducted not on a single-cell level, but at the “population level”. The average gene expression across sampled cells is obtained from bulk RNA-seq data, which masks cell heterogeneity [3]. Notably, in analyzing stem cells, circulating tumor cells, and other rare populations, the target cells may not account for a sufficient proportion of the bulk sample, due to incorrect sampling. Additionally, bulk techniques ignore the subtle but possibly physiologically significant variations between nearly identical cells. The limitations become more apparent in the studies of developmental biology. For example, for cell fate specification and cell differentiation—the main aspects of developmental process for multicellular organism—bulk RNA-seq approaches cannot distinguish the subtle differences of gene expression profiles between neighboring cells (either stem cell or differentiated cell). The single-cell analysis technique of single-cell RNA-sequencing (scRNA-seq) was therefore developed and has transformed our understanding of developmental biology for both humans and animals. Nonetheless, loss of spatial information during scRNA-seq severely limits our understanding of the specific functions of the cells performed by various spatial regions of the tissues or organs. Further application of spatial transcriptomics (ST) techniques has revealed the unbiased spatial organization of cell populations in the tissues or organs and assessed spatiotemporal gene expression dynamics during development, thereby improving the current understanding the developmental biology. In this review, we briefly summarize the application of scRNA-seq and ST techniques to developmental biology studies. We also discuss the status and limitations of current spatial transcriptomics, as well as future prospects.

## 2. scRNA-Seq Techniques and Developmental Biology

Since the first report in 2009 [4], scRNA-seq has been used extensively to study cellular heterogeneity at the single-cell level in tissues. The scRNA-seq technique, unlike bulk RNA-seq, allows transcriptome analysis at a single-cell resolution for tissues at various developmental stages, thereby representing the degree of cellular heterogeneity for transcriptome data accurately [5,6].

### 2.1. Single Cell Isolation Techniques Have Enabled scRNA-seq

In fact, RNA-seq at single-cell resolution can be obtained using scRNA-seq approaches by ‘simply’ doing single-cell isolation prior to RNA sequencing. Currently, the following methods are used to isolate single cells from tissues: fluorescence-activated cell sorting (FACS), magnetic-activated cell sorting (MACS), laser capture microdissection (LCM), micromanipulation, and microfluidics [7,8]. These techniques can be classified as high-throughput or low-throughput isolation methods as follows. Micromanipulation and LCM are low-throughput methods that manually isolate single cells under a microscope based on cell morphology and staining characteristics [9]. A straightforward and practical method for single cell isolation is manual cell selection by micromanipulation. It works very well for separating live cultures or embryonic cells. In most labs, manual cell selection is simple to do, but the throughput is low and trained professionals are required to handle the device [7,10]. LCM is mostly employed to isolate single cells from fixed tissue slices [11]. The key benefit of LCM is its speed while preserving accuracy. It offers a quick and reliable way to produce pure target cell populations from a variety of tissue preparations. The morphology of the collected cells is highly preserved, and no surrounding tissues are damaged. Additionally, LCM lessens human contact with the samples, lowering the possibility of contamination. The method’s main flaw is that it requires a visual microscopic examination to spot individual cells in complicated tissue. Thus, a technician with cell identification skills is required. Despite the above benefits, single cells isolated by LCM may suffer harm [12]. In addition to the above manual cell isolation techniques, FACS, MACS, and microfluidic techniques have been developed to separate single cells in an automated high-throughput way. FACS is a sophisticated form of flow cytometry that targets and isolates cell populations using fluorescent markers [13]. Based on cell surface markers, cells are detected and divided using FACS technology. Each cell has a distinct surface phenotype due to antigenic ligands such as proteins and carbohydrates, and particular antibodies linked to the cell surface antigens are used to target cells with those antigens. As the FACS technology is high-throughput and adaptable, it is currently widely used in clinical and research facilities. However, it has certain drawbacks, such as the need for a large number of suspended cells as a starting material and the inability to identify individual cells from small cell populations [7]. MACS is the other high-throughput cell isolation technique, which relies on cell surface antigens identified by antibodies or streptavidin-coated magnetic beads to isolate various cell types [14]. MACS can isolate cells with better than 90% purity, though it is expensive [15]. The use of high-throughput single-cell isolation techniques such as FACS and MACS for diagnostic purposes is restricted by their complexity, expense, and reagent requirements, which include labeling antibodies, magnetic nanoparticles, and sheath fluids. Another single-cell isolation technique, called microfluidics, bypasses the labeling method during single-cell sorting. Microfluidic techniques sort cells based on the inherent physical characteristics of cells, such as cell size, shape, density, deformability, electric polarizability/impedance, and other hydrodynamic features, as opposed to labeling tactics like FACS and MACS. Therefore, microfluidics is frequently utilized for single-cell separation. In particular, droplet-based microfluidics (also called microdroplets) is currently the most popular high-throughput platform; in microdroplets, single cells are masked by nanoliter droplets that contain a lysis buffer and barcoded beads using microfluidic and reverse emulsion devices. With its high throughput, low sample consumption, low analysis cost, and accurate fluid control, microfluidics has gained popularity, especially in mapping single-cell atlases for multicellular organisms [16,17]. Overall, these single-cell isolation techniques offer clear benefits, with large gains in capture effectiveness and target cell purity, further enabling RNA-seq at single-cell resolution.

### 2.2. Application of scRNA-seq in Developmental Biology Studies

Using the above single-cell isolation tools, scRNA-seq has enabled the most direct way to understand the embryonic developmental process, performing cell-by-cell transcriptome analysis. To begin with, scRNA-seq was used to study the early embryonic development of vertebrate animals at single-cell resolution. In 2009, scRNA-seq was for the first time used to study mammalian embryonic development in mice [4]; the transcriptome of a single mouse blastomere cell revealed 5270 genes, among which 1753 genes had not been detected by previous bulk approaches such as cDNA microarray. Since then, scRNA-seq has matured to the point where it can readily generate large single-cell atlases of developing mouse embryos [18]. The mechanism of mesodermal lineage diversification towards the hematopoietic system has previously been not studied because traditional bulk RNA-seq requires a large number of input cells, which could currently be addressed by scRNA-seq; sequencing 1205 single cells covering a time course from early gastrulation at embryonic day E6.5 to the generation of primitive red blood cells at E7.75 has revealed the first transcriptome-wide in vivo view of early mesoderm formation during mammalian gastrulation [19]. Cell fate specification during hematopoiesis could also be investigated using scRNA-seq in combination with the DNA barcode-based cell lineage tracing method [20]. Furthermore, a ‘global’ cell atlas of mouse organogenesis, essential for a comprehensive understanding of mammalian organogenesis, could be generated through scRNA-seq of approximately 2 million mouse embryo cells [21]. In zebrafish, scRNA-seq has generated transcriptome data of a large number of embryo cells, essential for high-throughput mapping of cellular differentiation hierarchies across the developmental trajectories during embryogenesis [22,23,24]. In monkeys, scRNA-seq of the lineage of ectoderm, trophectoderm, and primitive endoderm has revealed unique transcriptional programs and chromatin dynamics underlying monkey post-implantation development [25]. These scRNA-seq techniques have also been employed to study the embryo development of humans. As the number of cells in the early stages of embryonic development is limited, scRNA-seq has been employed to study embryonic development systematically; it could reveal dynamic gene expression during early embryonic development, differentiation, and reprogramming [26]. scRNA-seq of 124 cells from human preimplantation embryos and embryonic stem cells at different developmental stages has enabled the detection of 22,687 genes, many more than 9735 genes previously detected by cDNA microarray [26]. scRNA-seq of 1529 individual cells from 88 human preimplantation embryos has systematically elucidated the transcriptome map of the pre-implantation development of human embryos. The results revealed that early embryonic cells first undergo an intermediate state of co-expression of lineage-specific genes and then differentiate into trophectoderm, epiblast, and primitive endoderm lineages to form blastocysts [27]. Sequencing of over 8000 cells from 65 embryos before and after transplantation has been performed for systematic analysis of the implantation growth of fertilized eggs after artificial insemination in vitro, as well. These findings offer insights into the complex molecular mechanisms that regulate human embryo implantation, which can be exploited for in vitro derivation and directed differentiation of pluripotent stem cells [28]. In addition to embryo development, scRNA-seq has been extensively employed to study tissue differentiation and organ development. scRNA-seq for more than 2300 single cells of the human prefrontal cortex from 8 to 26 weeks of pregnancy has generated the single-cell transcriptome map for various cell types and developmental interactions during organogenesis [29]. scRNA-seq for 5227 single cells in four adult digestive tract organs (the esophagus, stomach, small intestine, and large intestine) and the embryonic large intestine between 6 and 25 weeks of gestation could elucidate their developmental processes, signaling pathways, cell cycles, metabolisms for nutrient absorption and digestion, and transcription factors [30]. Cell-by-cell transcriptome analysis through scRNA-seq for kidney cells could help to discover new IC-tran-PC cell types (AC19) in the adult kidney and unique forms of S-shaped body cells (FC1, FC9, and FC15) in the fetal kidney [31]. Taken together, the application of scRNA-seq techniques has enabled a generation of single-cell resolution transcriptome data for multicellular tissues or organs, as well as embryos, which help developmental biologists obtain better insights into the molecular mechanism of development.

## 3. ST Techniques and Developmental Biology

As described above, scRNA-seq has greatly advanced developmental biology studies. However, as scRNA-seq calls for single-cell isolation from solid tissues and organs, it is inevitable that isolated cells lose the information of the original tissue coordinates. For example, in scRNA-seq for kidney cells, individual cells of the kidney organ and its tissues are dissociated from their spatial composition and lose spatial information, otherwise they could not be further sequenced for single-cell resolution [32]. ST techniques now emerge to overcome this artefact by assaying cells in their native tissue environment. ST approaches enable developmental biologists to define the spatial distribution of mRNA molecules, offering critical insights in the disciplines of embryology, cancer, immunology, and histology. In multicellular organisms, the functioning of the individual cells can only be fully explained in the context of pinpointing their precise location within the body. ST strategies attempt to clarify the characteristics of cells in this way [33].

### 3.1. Classification of Ever-Emerged ST Techniques

To date, numerous ST techniques have been developed [34], which are based on either imaging or sequencing, in principle. They are practically classified into several categories, including in vitro capture and sequencing-, fluorescent in situ hybridization-, in situ barcoded amplification and sequencing-, and in situ capture and sequencing-based techniques (Figure 1).

In vitro capture and sequencing-based ST techniques depend on laser capture microdissection (LCM) [12,35] or other tissue dissection methods (such as manual slicing), allowing further RNA transcript profiling and cDNA library generation of the retrieved cells. This category typically includes RNA sequencing of individual cryosections [36], transcriptome in vivo analysis (TIVA) [37], RNA tomography (tomo-seq) [38,39], LCM coupled with Smart-Seq2 RNA sequencing (LCM-seq) [40,41], Geo-seq [42], NICHE-seq [43], polony (or DNA cluster)-indexed library-sequencing (PIXEL-seq) [44], and ProximID [45]. Fluorescent in situ hybridization (FISH)-based ST techniques rely on direct imaging of individual RNA molecules in single cells using multiple fluorophore-labeled probes. They include single-molecule FISH (smFISH) [46,47], RNAscope [48], sequential FISH (seqFISH) [49,50], multiplexed error-robust FISH (MERFISH) [51,52,53], single-molecule hybridization chain reaction (smHCR) [54], ouroboros smFISH (osmFISH) [55], extended sequential FISH (seqFISH+) [56], DNA microscopy [57], and so on. In situ barcoded amplification and sequencing-based techniques are the third category of ST techniques, which include in situ sequencing (ISS) using padlock probes [58], Fluorescent in situ sequencing (FISSEQ) [59,60], Barcode in situ targeted sequencing (Barista-seq) [61], barcoded oligonucleotides ligated on RNA amplified for multiplexed and parallel in situ analyses (BOLORAMIS) [62], expansion sequencing (ExSeq) [63], and Spatially-resolved transcript amplicon readout mapping (STARmap) [64]. In situ capture and sequencing-based techniques include Slide-seq [65], RNA sequencing using the peroxidase enzyme APEX2 (APEX-seq) [66], high-definition spatial transcriptomics (HDST) [67], deterministic barcoding in tissue for spatial omics sequencing (DBiT-seq) [68], sci-Space [69], 10X Visium [70], Seq-Scope [71], and spatial enhanced resolution omics-sequencing (Stereo-seq) [72,73]. In addition to the above categories, there are GeoMx [74] and photo-isolation chemistry-based transcriptome analysis [75], as well.

### 3.2. General Workflow of the ST Techniques

In spatial barcoding-dependent spatial transcriptomics approaches, a slice of tissue is placed onto the slide such that RNA from cells is tagged with the spatial barcodes. These barcodes provide information that enables RNA captured therein to be related to the original spatial position. These spatial barcodes can be created using randomly placed beads or a regular grid of spots. In addition to improving gene coverage and capture efficiency, succeeding strategies aim to create spatial barcodes in which the original place may be pinpointed with ever finer resolution. Visium technology uses an array of roughly 5000 spots with a diameter of 55 μm to capture gene expression [70]. Each spot’s expression information is recorded together with a stained image of the tissue. Slide-seq makes use of beads placed at random on a puck and has a 10 μm spatial resolution [65]. Seq-scope further improves the resolution with a center-to-center distance of 1 μm [71], while HDST captures at a resolution of 2 μm [67]. More recently, much finer resolution (200 nm) has been enabled by the Stereo-seq technique [72,73]. FISH techniques use finer-resolution optical imaging to identify specific RNA molecules. These techniques use imaging to quantify the fluorescent hues produced when RNA molecules hybridize, allowing for the identification and localization of RNA molecules. A variety of RNA, sorted exponentially in the number of rounds, can be identified by recording a certain RNA species as a sequence of colors and then labeling that RNA with the appropriate colors in subsequent imaging rounds. The RNA molecules are then sorted into groups according to the cell they came from to create a cell-by-count matrix that is spatially indexed by the location of each cell’s centroid. MERFISH can measure over 10,000 genes and uses error coding to improve measurement accuracy [51]. SeqFISH+ can cover up to 24,000 genes and uses more colors to cut down on the number of imaging rounds necessary for data collection [56]. FISH datasets typically capture fewer genes than spatial barcoding techniques but allow for precise localization of individual RNA molecules and a noticeably greater transcript capture depth, providing a more accurate image of the genes that are captured. In situ sequencing techniques like FISSEQ [59,60], BaristaSeq [61], and STARmap [64], in which RNA molecules are reverse transcribed into DNA and then sequenced inside the cell, are other ways to obtain the spatial transcriptome data. Alternatively, techniques utilizing cryosectioning, such as Geo-seq [42] and Tomo-seq [38,39], divide tissue into tiny slices before performing RNA sequencing. This enables non-spatial sequencing to be used for the final collection, increasing capture efficiency, but the number of spatial locations that can be retrieved and the resolution at which they can be separated is severely limited in those techniques.

### 3.3. Application of ST Techniques in Developmental Biology Studies

The above ST techniques have recently been applied to developmental biology studies, uncovering novel insights into the developmental process of organs and embryos in animals and humans. In 2018, FISH and immunohistochemistry have visualized global transcriptome ribosomal proteins of mouse oocyte and early embryo cells at a subcellular level, revealing unique and unexpected roles of translation machinery itself in directing essential aspects of oocyte and early embryo development [76]. Tomo-seq has enabled the establishment of a genome-wide expression dataset with finer spatial resolution for the developing zebrafish heart. Spatial mapping of the expression dataset for approximately 13,000 genes and over 1100 differentially expressed genes (DEGs) has uncovered spatially restricted Wnt/β-catenin signaling activity in pacemaker cells, which was controlled by Islet-1 activity, and also controlled heart rate by regulating the pacemaker cellular response to parasympathetic stimuli. This finer-resolution transcriptome map for embryonic heart cells has exposed a spatial view of molecular pathways important for specific cardiac functions [77]. In 2019, the gene expression landscape of human heart development was explored by ST techniques. A spatiotemporal overview of human heart development was initially obtained by immunohistochemical staining of tissue sections from human embryonic cardiac samples at various developmental stages. Then, the ST analysis was performed to output the spatial gene expression patterns (approximately 1700 genes and 3800 unique transcripts per spot) at spot-resolution. As a spot contained about 30 cells, scRNA-seq was further employed to deconvolve the gene expression heterogeneity for each cell in the spot gene expression dataset. These combined approaches have successfully explored global spatial transcriptional patterns in tissues [78]. Another study has employed combined approaches of Geo-seq and scRNA-seq to investigate molecular genealogy of cell fate specification and tissue organization in the early mouse embryo. Geo-seq analysis generated a spatiotemporal transcriptome of discrete cell populations of 5–40 cells, resulting in an unprecedented depth and quality catalog of transcripts in embryonic germ-layer tissues. scRNA-seq data has been used to deconvolve the spatial endoderm domain dataset to supplement their cell cluster information, finally obtaining a spatial transcriptome dataset of developing mouse embryos at a single-cell resolution [79]. Another combined approach of ST and scRNA-seq has been employed in mouse embryonic stem cell research. Through scRNA-seq analysis, 25,202 cells have been dissociated from 100 gastruloids at 120 h after aggregation, and their single-cell resolution transcriptome data were generated and then subjected to cell clustering. A total of 13 cell clusters were obtained. Cells in cluster 1–8 were determined to be ordered along neural and mesodermal differentiation trajectories through cluster gene expression analysis. Tomo-seq was subsequently employed to investigate neural and mesodermal differentiation trajectories, which have a strong spatial component. As a result, gene expression patterns in the gastruloids of mouse embryos have been spatially illustrated at a genome-wide scale [80]. In another example, the spatial-temporal developmental trajectories of mouse gut endoderm were delineated using scRNA-seq and the ISH-based ST technique. Using scRNA-seq, entire endoderm population cells (112,217 cells) have been dissociated and then sequenced. Further clustering and analysis for developmental trajectories have been validated by probe-dependent spatial transcriptomic analysis via ISH [81]. In 2020, the spatial overview of gastrulation of human embryonic stem cells was explained with the application of the tomo-seq technique [82]. Since 2021, there have been an increasing number of reports on the application of ST techniques, particularly lung organogenesis [83] and midbrain–hindbrain boundary patterning and gut tube development [84] and the development of various organs, such as the cerebral cortex [85,86], embryonic gastrointestinal tract [87], human embryonic liver [88], mesoderm [89], dorsal midbrain [72] in mice, the conceptus attachment during early placentation [90] in pigs, late-stage embryos and larvae development [91] in *Drosophila*, embryo development [92] in zebrafish, kidney [32], gonadal [93], and intestinal development [94] in human beings, and early gastrulation in utero [95] in monkeys (Table 1).

Before the application of ST techniques, scRNA-seq had already made significant progress in understanding the developmental biology of the above organs; however, the lack of spatial information for single cells limited our understanding of the specific functions of the cells performed by various spatial regions of the organs. The application of ST techniques has revealed the unbiased spatial organization of cell populations in the organs and assessed spatiotemporal gene expression dynamics during organ development (listed in Table 1).

For example, in understanding lung organogenesis, previously developed scRNA-seq and lineage-tracing techniques have provided extraordinarily detailed insights into the function of every single cell of a lung organ; however, cell locations and related fates in the whole lung organ have not yet been precisely explored. The application of ST, such as RNA-scope, has enabled spatial imaging developmental trajectories of various cell types such as epithelial, endothelial, and mesenchymal cells during lung organogenesis, thereby providing the integrated view of the cell fate specification during the later stages of lung development [83].

## 4. Limitations of Current Tools and Approaches for Spatial Transcriptomics in Developmental Biology Studies

### 4.1. Limitations of Current ST Tools

From Table 1, it is obvious that the ST techniques have mostly been employed in combination with scRNA-seq. ST techniques have emerged to address the issue of spatial context that is lost by cell dissociation in scRNA-seq practices. But except for the spatial interrogating ability, the other aspects—such as transcriptome resolution and transcript coverage, as well as transcript depth of the current ST methods—have not yet been matured to generate single-cell-resolution transcriptome data with as high-transcript coverage and depth as scRNA-seq. Thus, current ST methods still cannot completely replace scRNA-seq. For example, image-based ST techniques have been built on the basis of the single-molecule ISH technique. The transcript coverage degrees continue to improve to the current state: they can localize hundreds to thousands of genes in intact tissue but not genome-wide scale. Sequencing-based ST techniques have relatively finer resolution and genome-wide transcript coverage, but the transcript capture depth is not high enough, which results in a loss of information for genes with relatively low expression levels. In terms of transcript depth, the imaging-based ST techniques mostly employ probe-dependent targeted hybridization or amplification, thus revealing significantly higher transcript depth than sequencing-based techniques but lower transcript coverage. Moreover, spot resolutions of most sequencing-based ST techniques still need to be improved to reach smaller components than a single cell. Therefore, ST techniques still require technical improvement to reach single-cell resolution, genome-wide transcript coverage, and transcript capture depth as high as scRNA-seq techniques. Overall, it is obvious that the technical performance of currently available ST techniques is not yet satisfactory enough to generate the data with a resolution as fine as single-cell methods, albeit they have been emerged as the next-generation tools to complement the drawback of scRNA-seq, which does not preserve spatial information when generating data. In other words, though a previous-generation tool, scRNA-seq remains a major tool for spatial transcriptomic studies, although it is the single-cell method (Table 1, Figure 2), and integration of the data from both ST techniques and scRNA-seq is currently one of the most optimized spatial transcriptomic approaches. The efforts to integrate both techniques have been comprehensively reviewed in [96].

### 4.2. Limitations in Current Integrative Approaches for Spatial Transcriptomic Studies

Integration of current spatial transcriptome and scRNA-seq data, however, has a serious problem. The single-cell isolation methods in scRNA-seq analysis can generate errors in scRNA-seq data and, consequently, affect the final scientific conclusion drawn from the integrative approach of spatial transcriptomics and scRNA-seq. During the isolation of single cells from solid tissue, single cells undergo an in vitro culture and treatments, which are different from natural tissue conditions and, therefore, can seriously affect the intracellular transcriptome (Figure 3).

We examine the single-cell isolation methods adopted—in particular, the studies in Table 1. In one study, single oocyte cells were isolated away from mouse ovaries and transferred to an in vitro culture medium, with a chemical called 3-isobutyl-1 -methylxanthine to prevent oocyte meiosis [76]. Another study, which performed scRNA-seq for heart tissue, has obtained heart single cells by mincing, chilling at 4 °C (not fast-freezing), and subsequent trypsin and collagenase or laser treatments, as well as suspension culture [78]. More studies mentioned that they also have adopted similar isolation protocols or treatments [80,81,83,85,86,87,90,92,93]. All these studies did not mention whether physical or chemical isolation and further non-natural treatments could affect cell transcriptome or not. If the key DEGs identified in previous studies are the genes up- or down-regulated by induction from artificial treatments during single-cell isolation prior to scRNA-seq analyses, it cannot be concluded that the DEGs are associated with natural developmental processes. This questions the reliability of scRNA-seq data that have previously been generated by scRNA-seq analysis experiments and also affects currently adopted integrative approaches of scRNA-seq and ST.

However, the limitations of the single-cell isolation methods mentioned by previous studies are few; primarily mentioned is the loss of spatial information through the dissociation of tissue into single cells, which is a motivation for the development of ST techniques. Another one is the cell viability of single cells after dissociation from tissue. Nearly none of the previous studies have considered whether the single-cell transcriptome data could be affected by various artificial factors during the preparation of single cells prior to scRNA-seq. Only a few studies have briefly mentioned that the single-cell isolation step during tissue dissociation can alter gene expression and generate biased scRNA-seq data, but these did not mention any specific examples [97,98]. Also, no reports on the effects of physical stress or chemical treatments on the single-cell transcriptome have been made until now. Ideally, single-cell isolation protocols should not include any factors that can exert inducive and stress-like effects on intracellular transcriptome, thereby maintaining the transcriptome in its original state as before isolated. In terms of ST techniques, most of them do not need a single-cell isolation process, though they require tissue slicing at the macroscale. However, they seemed to consider the negative effects of tissue sample preparation operations on the transcriptome state, albeit only a few aspects; for instance, the tissue samples were stored at extremely low temperature, and the tissue spot capture operation was performed rapidly on the stored tissue samples immediately after quick-thawing [89]. Currently, it is impossible to obtain the control samples (single cell or tissue spot sample) with the transcriptome in its original state. Moreover, the bulk sample (entire tissue) cannot be used as a control due to bulk RNA-seq generating average transcriptome data. At present, no one has precisely demonstrated whether or not current single cell or tissue spot preparation protocols affect the cellular transcriptome. But further technical innovations in single cell or tissue spot preparation protocols will be made in the near future to generate original unaffected transcriptome data at single-cell resolution or finer. In situ capture and sequencing-based ST techniques, such as 10X Visium [70], Slide-seq [65], APEX-seq [66], HDST [67], sci-Space [69], seq-scope [71], and Stereo-seq [72,73] have enabled tissue dissection-free sequencing, thereby significantly reducing the risks from sample preparation as mentioned above; however, most of them have a resolution larger than a typical single cell. Only the recently emerged Stereo-seq technique has a very fine resolution smaller than single cell [72,73,92]. However, it must be further improved to detect much smaller (than a typical single cell size) cells such as immune cells [72].

## 5. Integration of Spatial Transcriptomics with Spatial Proteomics or Spatial Metabolomics: Future Prospect

### 5.1. Spatial Transcriptomics Is Just an Entry for Spatial Omics

The ST techniques generate spatial transcriptome data that can reveal essential spatial profiles of gene expression, and their application has greatly contributed to understanding developmental biology during the last decade, as shown in Table 1. Despite these advances, spatial transcriptomics is just an entry point for looking at the other spatial ‘omics’ sciences, such as spatial proteomics and spatial metabolomics. The basic flow of genetic information in a cell is as follows. The DNA is transcribed or copied into a form known as “RNA”. The complete set of RNA, also called its transcriptome, is subject to some editing (cutting and pasting) to become mRNA, which carries information to the ribosome, the protein factory of the cell, which then translates the message into protein, which then participates further in all cellular biological activities. Thus, transcriptome reveals only one aspect of the complex mechanism that keeps an organism running. Generation of the spatial transcriptome is only one step towards understanding the developmental process with the spatial view, which by itself does not specify everything that happens within the organism. Proteins, products translated from mRNAs, are the real effector molecules that are responsible for an endless number of tasks within the cell. Protein subcellular localization is tightly controlled and intimately linked to protein function in developmental biology. Therefore, understanding spatial localizations of proteins and their dynamics at the subcellular level is essential for a complete understanding of cell biology. Now, spatial proteomics techniques that are based on imaging-based and mass spectrometry (MS)- based approaches have become more accessible to developmental biologists. Studies on the human proteome are now beginning to demonstrate a complex architecture, which includes single-cell variations, dynamic protein translocations, altering interaction networks, and protein localization to different compartments. Additionally, comparative spatial proteomics has been successfully applied in multiple studies as a method for identifying disease processes. Spatial proteomics is entering an era of integration with developmental biology and medical research, allowing for an unbiased systems-level understanding of biological processes [99]. Likewise, the lack of spatial resolution in metabolomics is now a major challenge against current efforts to understand developmental biology. In a conventional metabolomics experiment, metabolites are extracted from homogenized tissues and then analyzed by LC-MS or NMR; thus, any information about the spatial distribution of the metabolites is lost, making it difficult for developmental biologists to precisely interrogate the position-specific biological roles of any of the detected molecules. The importance of localization is obvious in the case of proteins (for example, nuclear and cytosolic localization of a transcription factor may result in opposite phenotypes), and, similarly, different cellular and subcellular localization patterns of metabolites may be associated with significantly different biology [100]. The topic of developmental metabolism has been rarely mentioned or was even ignored since the introduction of modern molecular biology, although metabolic studies played a significant role in the early history of developmental biology research. However, metabolism has lately resurfaced as a focus of biomedical research, and, as a result, developmental biologists are once again investigating the metabolic processes that influence growth, development, and maturation at spatial resolution.

### 5.2. Spatial Multi-Omics Data Would Provide Novel and Comprehensive Insights into Developmental Biology

Although rapid advances in individual techniques of spatial transcriptomics, spatial proteomics, and spatial metabolomics have recently been made, these techniques alone cannot capture the entire biological complexity of the various developmental processes in animal and humans. Integrated spatial omics, which combines spatial transcriptomics, spatial proteomics, and spatial metabolomics, has therefore emerged as a solution to provide comprehensive knowledge of developmental biology. However, at present, integrative spatial omics approaches have not been widely adopted for developmental biology studies. There are only a small number of publications on the application of integrative spatial omics approaches to research, in which the integration methods have not yet been fully optimized. In 2018, the ST technique APEX-seq was employed, along with a spatial proteomics tool, APEX-mass spectrometry (APEX-MS), to uncover the intracellular activities of ribonucleoprotein complexes. Ribonucleoprotein is the complex of RNA and protein, and, therefore, the integrative approach of spatial transcriptomics and spatial proteomics could unveil more comprehensive and detailed knowledge of intracellular ribonucleoprotein activities. APEX-seq, in conjunction with proteomics, has provided new insights into the organization of translation initiation complexes on active mRNAs and the composition of repressive RNA granules, which could not be obtained by a solitary approach of only the spatial transcriptomics [101]. In 2021, the ST technique was used in combination with a spatial proteomics tool to assess tissue samples from 25 patients with high-grade urothelial MIBC (muscle-invasive bladder cancer) treated with surgery alone. Different cell types and compositions in tumors were identified; notably, a second-generation bladder subtype architecture was defined, which could guide cancer therapy more precisely and effectively by improving therapeutic response prediction ability. In the study, the integration mode of spatial transcriptomics and spatial proteomics was that spatial transcriptome data analysis was the major work and spatial proteomic data was used to validate the reliability of spatial transcriptome data [102]. Integration of spatial metabolomics and spatial transcriptomics has also been reported, in which the spatial metabolome data analysis was the major work and was validated by spatial transcriptome data. In the report, the metabolites of the human fibrotic liver were analyzed by a new spatial metabolomics method called spatial single nuclear metabolomics (*SEAM*). The spatial metabolome data, which was obtained by the SEAM method, was further validated by Geo-seq (an ST method) [103]. Although it currently seems that the integration mode of different omics data has not yet been fully developed, further efforts to fully exploit individual omics tools or their combination will be made to provide better and more comprehensive insights into developmental biology.

## 6. Conclusions

In animals and humans, the developmental process from one cell to multicellular tissues and organs is not merely the exact replication process, which is called differentiation. Difference in gene expression profiles between individual cells throughout tissues and organs has long been the primary subject for investigating the mechanism of differentiation. Rapid advances in methodologies such as genomics tools have enabled large-scale genome-wide investigation for developmental processes. In particular, it was a breathtaking breakthrough when scRNA-seq techniques enabled the generation of the transcriptome at single-cell resolution. Cell heterogeneity between differentiated cells and stem cells, which has been masked in conventional bulk RNA-seq, can be explored in scRNA-seq, thus making it possible for scientists to better understand the mystery of differentiation and cell lineage during the developmental process. But scRNA-seq requires a single-cell isolation step prior to sequencing single cells, which results in the loss of spatial information of individual single cells and, consequently, limits our understanding of the specific functions of the cells performed by various spatial regions of the tissues or organs. This greatly promotes the emergence of spatial transcriptomics, but current ST techniques have been in the way of development. Thus, integration of ST and scRNA-seq analysis has currently been the most common approach (while the resolution of ST techniques continues to be finer) and has thus far made great advances in understanding developmental biology at spatial resolution. Despite these advances, current ST and scRNA-seq techniques seem to have serious problems in the preparation of samples for sequencing such as single cells or tissue spots. No one has ever considered whether the various stressing factors or treatments involved in sample preparation protocols could make significant changes in the original transcriptome state of single cells or tissue spots. If sample preparation protocols really affect the original transcriptome state, most of the results from previous scRNA-seq or spatial transcriptomics studies become unreliable. More significant innovations in sample preparation are needed to preserve the original transcriptome state of samples prior to sequencing. In addition, spatial transcriptomics is just an entry point for looking at the other spatial ‘omics’, sciences such as spatial proteomics and spatial metabolomics, and, thus, should be integrated with spatial proteomics or/and spatial metabolomics to provide better and more comprehensive insights into developmental biology.

## Figures and Tables

**Figure 1 biomolecules-13-00156-f001:**
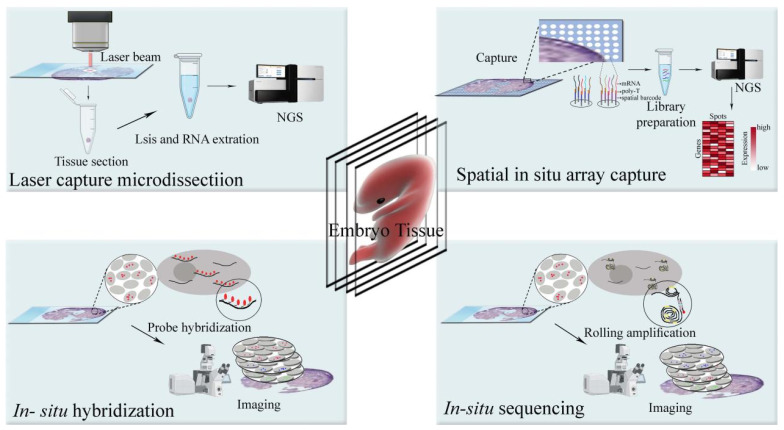
Schematic diagram of common ST techniques. There are two main categories of ST techniques: imaging and sequencing-based. Sequencing-based ST techniques are further classified into LCM-based or 10X Visium spatial sequencing methods. LCM-based spatial sequencing employs LCM dissection to capture the microsection of interest to be further in vitro sequenced. The 10X Visium method utilizes a spatially barcoded sequencing platform. The tissue sample is placed onto the platform without dissection and then in situ sequenced. Imaging-based ST techniques include fluorescent in situ hybridization- and in situ barcoded amplification-based methods. Fluorescent in situ hybridization-based ST techniques rely on direct imaging of individual RNA molecules in single cells using multiple fluorophore-labeled probes. In situ barcoded amplification-based methods are also based on in situ hybridization, but they use a distinct probe called padlock to amplify the hybridization signal through rolling amplification, thereby enhancing transcript capture depth.

**Figure 2 biomolecules-13-00156-f002:**
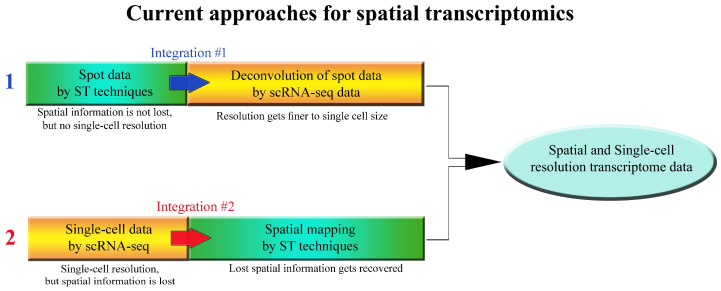
Integrative approaches for current spatial transcriptomics for developmental biology studies. There are mainly two ways to integrate both techniques: (1) spatial transcriptome data is generated at spot resolution, and then the spot data is further deconvolved into single-cell data using scRNA-seq data, and (2) single-cell transcriptome data is generated by scRNA-seq and then lost spatial context is recovered through spatial mapping of scRNA-seq data by ST techniques.

**Figure 3 biomolecules-13-00156-f003:**
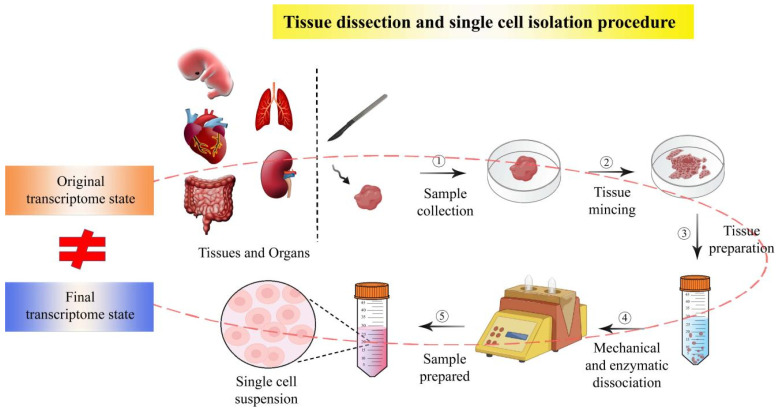
Limitation of the sample preparation protocol for current spatial transcriptomic studies. During the isolation of single cells from solid tissue, single cells undergo in vitro culture and treatments, which are different from natural tissue conditions, and their original transcriptome state, therefore, can be seriously affected. Single cells are isolated away from tissues or organs and transferred to in vitro culture medium with various chemicals and enzymes. The mechanical or enzymatic isolation and further non-natural treatments could change cell transcriptome state, consequently affecting the quality of transcriptomes data obtained from currently adopted integrative approaches of scRNA-seq and ST.

**Table 1 biomolecules-13-00156-t001:** Current advances in spatial transcriptomic studies in developmental biology.

Organism	Sequenced Samples	Sample Preparation	Sequencing Techniques	How to Obtain Single Cell- and Spatial-Resolution Transcriptome Data	References
human	heart tissues at 4.5~9 PCW single cells	tissue dissection tissue dissociation cell suspension	10× Visium scRNA-seq ISS	1. Generation of spatial transcriptome data at spot (~equal to size of 30 cells) resolution by 10× Visium 2. Deconvolution of spot transcriptome data into single-cell-resolution with scRNA-seq data 3. Validation of overall spatial transcriptome data by ISS	[78]
	intestine tissues at 8~22 PCW single cells	full tissue digestion tissue block sectioning	scRNA-seq 10× Visium	1. Generation of single-cell-resolution transcriptome data by scRNA-seq 2. Generation of spatial transcriptome data by 10× Visium and spatial mapping of scRNA-seq data with 10× Visium data	[94]
	gastruloids	chiron pre-treatment cryo-sectioning	Tomo-seq	Generation of spatial transcriptome data at each section (20-μm sections) by Tomo-seq	[82]
	gonadal tissue at 6~21 PCW single cells	tissue dissection tissue dissociation cell suspension cryo-sectioning paraformaldehyde fixing	scRNA-seq smFISH 10× Visium	1. Generation of single-cell-resolution transcriptome data by scRNA-seq 2. Providing spatial information for partial genes by probe-dependent smFISH 3. Generation of spatial transcriptome data by 10× Visium and spatial mapping of scRNA-seq data with 10× Visium data	[93]
	liver tissue at 8~17 PCW	cryo-sectioning tissue dissection	10× Visium	1. Generation of spatial transcriptome data at spot (100μm) resolution by 10× Visium 2. Deconvolution of spot data into single-cell resolution with previous scRNA-seq data	[88]
	kidney tissue at 9~18 PCW single cells	cryo-sectioning cell suspension	10× Visium scRNA-seq	1. Generation of spatial transcriptome data at spot (100μm) resolution by 10× Visium 2. Deconvolution of spot transcriptome data into single-cell-resolution with scRNA-seq data	[32]
mouse	lung tissues at E12~P14 single cells	tissue dissection tissue dissociation cell suspension	scRNA-seq RNAScope	1. Generation of single-cell resolution transcriptome data by scRNA-seq 2. Providing spatial information for partial genes of scRNA-seq data by probe-dependent RNA Scope	[83]
	embryo tissues at E2.5~E7.5 single cells	1. cryo-sectioning tissue dissection tissue dissociation 2. manually cell picking	Geo-seq scRNAseq	1. Generation of spatial transcriptome data at capture area (5–40 cells) by Geo-seq 2. Deconvolution of the Geo-seq data into single-cell resolution using scRNA-seq data	[79]
	somatosensory cortex tissues at E10.5~E18.5 and P1~P4 single cells	tissue dissection tissue dissociation cell suspension tissue block sectioning	scRNA-seq Slide-seq	1. Generation of single-cell resolution transcriptome data by scRNA-seq 2. Spatial mapping of scRNA-seq data onto Slide-seq data with Tangram	[85]
	oocyte and 2-cell embryo	IBMX treatment oocytes picking paraformaldehyde fixing permeabilizing in Triton X-100	smRNA FISH RCA FISH	1. Detecting RNA localization by smRNA FISH 2. Visualizing the whole cellular transcriptome by RCA FISH	[76]
	Embryo tissues at E14.0	cryo-sectioning	sci-Space	Generation of spatial transcriptome data at spot (200 μm) resolution by sci-Space	[69]
	embryo tissues at E14.5	cryo-sectioning	PIC RNA-seq	Generation of spatial transcriptome data at regions of interest by PIC RNA-seq	[75]
	embryo tissues at E8.5~E8.75 single cells	tissue dissection paraformaldehyde fixing cryo-sectioning	seqFISH scRNA-seq	1. Obtaining spatial information for partial genes by probe-dependent seqFISH 2. Spatial mapping of scRNA-seq data onto seqFISH data	[84]
	gut endoderm tissue at E3.5~E8.75 single cells	tissue dissection tissue dissociation cell suspension paraformaldehyde fixing	scRNA-seq ISH	1. Generation of single-cell resolution transcriptome data by scRNA-seq 2. Validation of cell cluster- and position-specific expression profile data by ISH	[81]
	gastruloids tissue at 120 h after aggregation Embryo tissues at E8.5~E9.5	cryo-sectioning	tomo-seq	Generation of transcriptome data at each section (8-μm and 20-μm sections) by Tomo-seq	[80]
	cerebral cortex tissue at E12.5 single cells	tissue dissociation cell suspension paraformaldehyde fixing	scRNA-seq ISH	1. Generation of single-cell resolution transcriptome data by scRNA-seq 2. Providing spatial information for partial genes of scRNA-seq data by probe-dependent ISH	[86]
	stomach and intestine tissues at E9.5~E15.5 single cells	tissue dissection tissue dissociation cell suspension cryo-sectioning	scRNA-seq 10× Visium,	1. Generation of single-cell resolution transcriptome data by scRNA-seq 2. Mapping spatial distributions of scRNA-seq data by 10× Visium	[87]
	embryo tissues at E9.5~E16.5 single cells	cryo-sectioning	Stereo-seq ISH scRNA-seq	1. Generation of spatial transcriptome data at spot (220 nm) resolution by Stereo-seq 2. Validation of spatial transcriptome data by ISH 3. Spatial alignment of scRNA-seq data with Stereo-seq by Tangram	[72]
	embryo tissues at E7.5	cryo-sectioning	LCM-seq	Generation of spatial transcriptome data at capture area (50–300 cells) by LCM-seq	[89]
pig	uterine tissue at G12~G15	formalin fixing cryo-sectioning	LCM-seq	Generation of spatial transcriptome data at capture area by LCM-seq	[90]
Marm-oset	preimplantation and postimplantation embryos (E15~E25) uterine tissue	cryo-sectioning cell picking by LCM	LCM-seq	Generation of spatial transcriptome data at capture area (1–3 cells) by LCM-seq	[95]
Zebra-fish	heart tissue at 2 dpf	cryo-sectioning	Tomo-seq	Generation of transcriptome data at each section (10-μm sections) by Tomo-seq	[77]
	embryo tissues at 3.3~24 hpf single cells	cryo-sectioning paraformaldehyde fixing tissue dissociation cell suspension	Stereo-seq ISH scRNA-seq	1. Generation of spatial transcriptome data at spot (220 nm) resolution by Stereo-seq 2. Obtaining spatial information for partial genes by probe-dependent ISH 3. Construction of single-cell- and spatial-resolution developmental trajectory by integrating scRNA-seq and Stereo-seq data	[92]
Droso-phila	embryo tissues at 14–16 ELh, 14–18 E, and 1–3 L	cryo-sectioning	Stereo-seq ISH	1. Generation of spatial transcriptome data at spot (220 nm) resolution by Stereo-seq 2. Validation of spatial transcriptome data by ISH	[91]

Abbreviations; dpf: days post-fertilization, E: embryonic day, ELh: egg laying hour, G: gestational day, hpf: hours post-fertilization, L: larvae day, LCM: laser capture microdissection, P: postnatal day, PCW: post-conception weeks, PIC: photo-isolation chemistry, IBXM: 3-isobutyl-1-methylxanthine.

## Data Availability

This study did not generate any unique datasets or code.

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
