# Peer review of "Advances and Challenges in Spatial Transcriptomics for Developmental Biology"

_biomolecules, 2023, doi:10.3390/biom13010156_

Round 1

Reviewer 1 Report

The manuscript by Choe et al. reviews the trends and limitations of single cell RNA-seq and spatial transcriptomic approaches that have been widely used recently.

Overall, it is a very well-organized review, and it is written to help you understand the trends in spatial transcriptomics by giving examples of developmental biology based on published papers. In particular, by introducing various spatial transcriptome approaches, it will be of great help to researchers as they can learn information about which experimental methods are being used and identify which areas have been improved during the development process.

However, the description of the limitations of current tools and approaches for spatial transcriptomics in developmental biology studies is already known as a problem in single cell experiments rather than about the limitations of the spatial transcriptome, and even Fig 3 shows a single cell research method rather than an explanation of spatial transcriptomic studies. For this part, authors should add examples of the limitations of spatial transcriptomic studies only, excluding the limitations of single cell approaches.

And there is no need to repeat what has already been written in the manuscript for figure legends ( Line 199~200, 375~382, 397~401). Also, in Table 1, it is necessary to adjust the table position and add lines to clearly separate the classifications. Finally, please correct minor typos.

Author Response

Response to reviewer 1 comments

Comments to the Author

The manuscript by Choe et al. reviews the trends and limitations of single cell RNA-seq and spatial transcriptomic approaches that have been widely used recently.

Overall, it is a very well-organized review, and it is written to help you understand the trends in spatial transcriptomics by giving examples of developmental biology based on published papers. In particular, by introducing various spatial transcriptome approaches, it will be of great help to researchers as they can learn information about which experimental methods are being used and identify which areas have been improved during the development process.

point 1: However, the description of the limitations of current tools and approaches for spatial transcriptomics in developmental biology studies is already known as a problem in single cell experiments rather than about the limitations of the spatial transcriptome, and even Fig 3 shows a single cell research method rather than an explanation of spatial transcriptomic studies. For this part, authors should add examples of the limitations of spatial transcriptomic studies only, excluding the limitations of single cell approaches.

Response1: We thank the reviewer for this comment. We appreciate the reviewer’s insightful suggestion and agree that the text and figure 3 about section for ‘limitation of current spatial transcriptomics’ were prepared in an unreasonable way. Actually, we think that this part is about the single cell techniques, too. However, we could not help to mention the limitation of single cell method in the section related to ‘spatial transcriptomics’. As mentioned in Abstract, loss of spatial information of individual cells encourages the emergence of spatial transcriptomic discipline and tools. But technical limitations of recently emerged spatial transcriptomic tools make them not able to completely replace the single cell methods (as described in section 4.1) and scRNA-seq is still the major tool for current spatial transcriptomics (as shown in Table 1 and Fig 2). More importantly, the limitation of single cell method seems to affect the quality of current spatial transcriptomic works more severely than limitation of spatial transcriptomic tools’ own as described in section 4.2 (this section has not been mentioned in previous studies, we for the first time describe it). Therefore, we think that limitation of single cell methods (described in section 4.2) makes itself indispensable to the section for limitations of current spatial transcriptomics. We apologize if we did not clearly confer this meaning in section 4. After careful consideration of the reviewer’s comment, we have revised some part of the section 4.1 and header of section 4.2 and hope that our idea is now clearly presented. Revision has been made as follows:

Line 377-380:

We deleted “Nonetheless, scientists now have integrated the data from both techniques (ST tech-niques and scRNA-seq) to complement the limitation of current ST techniques (Table 1, Figure 2), and the efforts to integrate both techniques have been comprehensively re-viewed in [96]”.

Then, we added “Overall, it is obvious that technical performance of currently available ST techniques is not yet satisfactory enough to generate the data with a resolution as fine as single-cell methods, albeit they have been emerged as the next-generation tools to complement the drawback of scRNA-seq which does not preserve spatial information when generating data. In other words, scRNA-seq as a previous generation tool still remains as a major tool for spatial transcriptomic studies although it is the single cell method (Table 1, Fig 2) and integration of the data from both ST techiniques and scRNA-seq is currently being one of the most optimized spatial transcriptomic approaches. The efforts to integrate both techniques have been comprehensively re-viewed in [96]. ”

  Line 393:

    Original header of section 4.2 sounds like we describe ‘only the spatial method’. We have changed the “4.2. Limitations of current spatial transcriptomic approaches” to “4.2. Limitations in current integrative approaches for spatial transcriptomic studies”

Point 2: And there is no need to repeat what has already been written in the manuscript for figure legends (Line 199~200, 375~382, 397~401). Also, in Table 1, it is necessary to adjust the table position and add lines to clearly separate the classifications. Finally, please correct minor typos.

Response2: We thank the reviewer for pointing this out. We agree and updated the figure legends. Repetitive parts of figure legends have been revised as follows:

Line 199-200: We have replaced original text with “There are two main categories of ST techniques such as imaging- or sequencing-based ST techniques.”

Line 375-382: We have deleted “ST techniques have been emerged to address the issue for spatial context that is lost by cell dis-sociation in scRNA-seq practices. But except the spatial interrogating ability, the other aspects such as transcriptome resolution and transcript coverage as well as transcript depth of the current ST techniques have not yet been matured to generate single-cell resolution transcriptome with high transcript coverage and depth as scRNA-seq. Thus, current ST methods still cannot com-pletely replace scRNA-seq. Nonetheless, ST techniques and scRNA-seq are currently used in combination to complement the limitation of current ST techniques (Table 1).”

  Line 397-401: We have deleted “If these artificial treatments are responsible for differential expression of key DEGs which have been identified in previous studies, the DEGs cannot be used to conclude that they play significant roles in natural developmental processes. This questions the reliability of scRNA-seq data that have previously been generated by scRNA-seq analysis experiments and” and revised “affects currently adopted integrative approaches of scRNA-seq and ST” as “affecting the quality of transcriptomes data obtained from currently adopted integrative approaches of scRNA-seq and ST”.

We also adjusted the Table 1 according to the reviewer and corrected several typos.

All the revisions have been tracked by “Track Changes” function of MS Word.

Reviewer 2 Report

The review paper “Advances and challenges in Spatial transcriptomics for developmental biology” by Choe et al. summarizes current technologies that examine gene expression at the single cell and spatial levels. To date, a number of cutting-edge tools have been developed that focus on the localization of RNA transcripts in single cells within a tissue. These are essential tools for examining and understanding processes in animal development. In addition, they discuss the limitations of current spatial transcriptomic tools and approaches. The authors provide a timely review paper on the various gene expression analyses, i.e. transcriptomics, utilized in developmental biology research. Three figures and one table are included in this review which effectively complement the information provided in the text. Overall, this is a well written and concise review paper containing pertinent information on the advances scientists have made to understand gene expression patterns using single cell and spatial transcriptomics.

Minor concerns:

1. The authors should carefully check their grammar and spelling throughout the manuscript. Several errors were noted. Here are just a few examples:

Abstract (line 14) – “exact replication” should be clarified. The authors might consider rephrasing to “exact replication of cells” so that replication is intended to mean that “many copies of the same cells” and not replication of some other context.

Abstract (line 19-20) – “sequencing transcriptome” should be changed to “sequencing transcriptomes”

Line 93 – “Despite above benefits” changed to “Despite the above benefits”

Line 126 – “Using above single cell” changed to “Using the above single cell”

Line 412 – “Other study” changed to “Another study”

Line 539 – “In animal and human being” changed to “In animals and humans”

2. Section 3, ST techniques (line 181) – The authors should briefly provide in the paragraph an example or two (with citations) where spatial location of cells within a solid tissue or organ might have “lost” information when isolated from the tissue.

Author Response

Response to reviewer 2 comments

Comments to the Author

The review paper “Advances and challenges in Spatial transcriptomics for developmental biology” by Choe et al. summarizes current technologies that examine gene expression at the single cell and spatial levels. To date, a number of cutting-edge tools have been developed that focus on the localization of RNA transcripts in single cells within a tissue. These are essential tools for examining and understanding processes in animal development. In addition, they discuss the limitations of current spatial transcriptomic tools and approaches. The authors provide a timely review paper on the various gene expression analyses, i.e. transcriptomics, utilized in developmental biology research. Three figures and one table are included in this review which effectively complement the information provided in the text. Overall, this is a well written and concise review paper containing pertinent information on the advances scientists have made to understand gene expression patterns using single cell and spatial transcriptomics.

Point 1: The authors should carefully check their grammar and spelling throughout the manuscript. Several errors were noted. Here are just a few examples:

  • Abstract (line 14) – “exact replication” should be clarified. The authors might consider rephrasing to “exact replication of cells” so that replication is intended to mean that “many copies of the same cells” and not replication of some other context.
  • Abstract (line 19-20) – “sequencing transcriptome” should be changed to “sequencing transcriptomes”
  • Line 93 – “Despite above benefits” changed to “Despite the above benefits”
  • Line 126 – “Using above single cell” changed to “Using the above single cell”
  • Line 412 – “Other study” changed to “Another study”
  • Line 539 – “In animal and human being” changed to “In animals and humans”

Response 1: We thank the reviewer for this comment. We have corrected the errors according to the reviewer’s suggestions. Revision has been made as follows:

  • Line 14: “exact replication” to “exact replication of cells”
  • line 19-20: “sequencing transcriptome” to “sequencing transcriptomes”
  • Line 93: “Despite above benefits” to “Despite the above benefits”
  • Line 126: “Using above single cell” to “Using the above single cell”
  • Line 412: “Other study” to “Another study”
  • Line 539 – “In animal and human being” to “In animals and humans”

In addition, we have carefully checked the manuscript and made several minor modifications which have been tracked by MS Word ‘track changes’ function.

Point 2: Section 3, ST techniques (line 181) – The authors should briefly provide in the paragraph an example or two (with citations) where spatial location of cells within a solid tissue or organ might have “lost” information when isolated from the tissue.

Response 2: We thank the reviewer for pointing this out. We agree and have supplemented some examples to the paragraph as follows:

Line 184-187: “For example, in scRNA-seq for kidney cells, individual cells of kidney organ and its tissues should be dissociated from their spatial composition, they lost spatial information otherwise they could not be further sequenced at single cell-resolution [92].”
